# Effect of Multi-Walled Carbon Nanotubes-Based Nanofluids on Marine Gas Turbine Intercooler Performance

**DOI:** 10.3390/nano11092300

**Published:** 2021-09-04

**Authors:** Salah Almurtaji, Naser Ali, Joao A. Teixeira, Abdulmajid Addali

**Affiliations:** 1School of Aerospace, Transport and Manufacturing (SATM), Cranfield University, Cranfield MK43 0AL, UK; j.a.amaral.teixeira@cranfield.ac.uk (J.A.T.); a.addali@cranfield.ac.uk (A.A.); 2Kuwait Army, Kuwait Ministry of Defense, Safat 13128, Kuwait; 3Kuwait Institute for Scientific Research, Energy and Building Research Center, Nanotechnology and Applications Program, Safat 13109, Kuwait; nmali@kisr.edu.kw

**Keywords:** colloidal, suspension, WR-21, MWCNT, intercooler effectiveness, pumping power

## Abstract

Coolants play a major role in the performance of heat exchanging systems. In a marine gas turbine engine, an intercooler is used to reduce the compressed gas temperature between the compressor stages. The thermophysical properties of the coolant running within the intercooler directly influence the level of enhancement in the performance of the unit. Therefore, employing working fluids of exceptional thermal properties is beneficial for improving performance in such applications, compared to conventional fluids. This paper investigates the effect of utilizing nanofluids for enhancing the performance of a marine gas turbine intercooler. Multi-walled carbon nanotubes (MWCNTs)-water with nanofluids at 0.01–0.10 vol % concentration were produced using a two-step controlled-temperature approach ranging from 10 °C to 50 °C. Next, the thermophysical properties of the as-prepared suspensions, such as density, thermal conductivity, specific heat capacity, and viscosity, were characterized. The intercooler performance was then determined by employing the measured data of the MWCNTs-based nanofluids thermophysical properties in theoretical formulae. This includes determining the intercooler effectiveness, heat transfer rate, gas outlet temperature, coolant outlet temperature, and pumping power. Finally, a comparison between a copper-based nanofluid from the literature with the as-prepared MWCNTs-based nanofluid was performed to determine the influence of each of these suspensions on the intercooler performance.

## 1. Introduction

For many years, scientists have attempted to improve the thermal performance of heat transfer devices to lower their overall energy consumption and/or reduce their constructional scale. These devices have a primary role in our daily lives, where they exist in a range of sizes from those in a domestic setting (e.g., portable heaters) up to large industrial scale heat exchangers (HEs) (e.g., power stations) [1]. Most of the initial attempts undertaken by the experts to improve these systems were focused on geometrical modifications, such as using plate HEs, tubular HEs, extended surface HEs and regenerative HEs as well as altering the flow arrangement of the working fluids (e.g., parallel flow, crossflow, and counterflow arrangements) and pass arrangements (i.e., one- or multi-pass) [2]. However, all of these previous methods have come to the point were limited enhancement in the thermal performance can be gained, and as such the focus of researchers today has shifted towards advancing the working fluid itself (i.e., exploring innovative fluids of higher thermal properties) [3]. Nanofluids are strong candidates due to their significant thermal properties compared to conventional liquids. This is because nanofluids are formed by dispersing nanoscale particles in a non-dissolving host liquid. For heat transfer applications, these particles have orders of magnitude higher thermal conductivity than their hosting environment and consequently increase the overall thermal conductivity of the mixture [4]. Historically, this type of advanced suspension was first discovered by Masuda et al. [5], in 1993, and later, defined by Choi and Eastman [6] in 1995. Despite the success of nanofluids in improving the thermal performance of many heat transfer systems [7,8,9], there are several applications that are yet to be adequately explored. One of these applications is related to the marine industry, specifically the intercooler unit of the WR-21 gas turbine engine. The WR-21 marine engine was designed and constructed by Westinghouse and Rolls-Royce, in 1991 [10], to power up the Royal Navy type 45 Air Defence Destroyer [11]. The unique feature that is associated with this engine is that it contains an intercooler unit, which is a type of heat exchanger, that is integrated between the compressor stages. The role of the intercooler is to reduce the compressed air temperature between the stages, and hence raise the thermal efficiency of the engine [12]. According to the latest literature survey that was conducted by Alsayegh and Ali [2], there are very few research works that investigate the use of nanofluids in similar systems. For instance, Zhao et al. [13,14] have theoretically investigated the effect of using Al_2_O_3_–water and Cu–water suspensions instead of water in the intercooler unit of the marine gas turbine. The researchers implemented the effectiveness-number of transfer units method to predict the flow and thermal transport performance of the system. The concentration of nanoparticles used was between 1.0 volume percentage (vol %) to 5.0 vol % for both nanofluids. In terms of the thermophysical properties of the suspensions, they were calculated using pre-existing formulas from the literature. Furthermore, the effect of primary parameters such as particle concentration, inlet working fluid temperature, Reynolds number (*Re*), and the other factors that are related to the gaseous fluid of the intercooler were all explored in terms of flow performance and heat transfer of the heat exchanging unit. The results revealed that all suspensions had enhanced the heat transfer performance over that of the conventional working fluid. Zhao et al. [15] have studied the operational conditions and constructional material influence in an intercooler in a previous study [13,14]. Three intercooler materials were investigated, namely, aluminium, copper, and copper–nickel alloy. The researchers found that using a copper–nickel alloy intercooler as well as raising the operating load from ~20% to 100% resulted in an increase in the compressed gas outlet temperature by ~13.5 °C and ~13.0 °C when employing variable properties and constant properties, respectively. As for the coolant temperature, an increase of ~7 °C was observed when changing the operating load from 20% to 100% regardless of using constant or variable liquid properties. Therefore, it can be concluded from their work that the parameters of the compressed gas have high sensitivity to the operating conditions compared to the coolant. It is important to note that the authors have noticed that the aluminium-based intercooler had a relatively balanced dynamic response between the compressed gas and coolant in comparison to the other two materials. Chintala et al. [16] experimented on a two-stage air compressor integrated with a counterflow intercooler unit to determine the changes in the system heat transfer behaviour when using Al_2_O_3_–water nanofluids. The examined suspensions were produced by sonicating 0.5–1.0 vol % of Al_2_O_3_, for 90 min and at 40 °C. Their results showed a maximum increase of 36.1% in the intercooler thermal efficiency when using the 1.0 vol % nanofluid at 4 bar compressor load. 

In previous literature [13,14,15,16] the studied nanofluids comprised dispersed nanoparticles of very low thermal conductivity (i.e., 40 W/m∙K and 401 W/m∙K for Al_2_O_3_ and Cu at room temperature, respectively) compared to todays advanced nanomaterials (e.g., ~4000 W/m∙K for carbon nanotubes (CNTs) at room temperature) [17]. Furthermore, the approach used in preparing nanofluids has a crucial effect on the suspension’s thermophysical properties. This means that parameters such as sonication duration, mixing intensity, base fluid temperature, particle shape, and concentration, would cause the resulting thermophysical properties (e.g., thermal conductivity, viscosity, density, specific heat capacity) of the mixture to vary [18,19]. Such high sensitivity in the nanofluid’s properties was frequently reported by scientists in the field [20,21,22], and many attempts were made to solve this issue [23].

In the light of the aforementioned, this paper will investigate the effect of preparation parameters of multi-walled CNTs (MWCNTs)–water nanofluids on the performance of a WR-21 intercooler. The current study is divided into two parts. The first stage deals with determining the changes in the thermophysical properties of the as-produced MWCNTs-based suspensions and the second stage focuses on the modelling of the intercooler thermal performance. The nanofluids were prepared through the two-step controlled-temperature approach (from 10 °C to 50 °C) using 0.01–0.1 vol % of MWCNTs, and sodium dodecyl sulfate (SDS) surfactant (1:1 weight ratio of SDS to MWCNTs) to improve the dispersion stability [24,25]. Afterwards, the thermophysical properties of the as-fabricated suspensions and physical stability were determined. Next, the obtained properties from the different examined samples were used in mathematical models to evaluate the thermal performance of the intercooler system. The output of this work is believed to be beneficial for researchers who are interested in developing the next generation of gas turbine intercoolers as well as for major industrial companies such as Rolls-Royce and Westinghouse.

## 2. Materials and Methods

### 2.1. Starting Materials and Equipment

The commercial MWCNT powder was obtained from SkySpring Nanomaterials Inc. (Houston, TX, USA) with purity of >90 wt %, outer diameter of 30–50 nm, inner diameter of 5–10 nm, and axial length of 5–15 μm. According to the manufacturer, the carbon-based nanomaterial was produced through a catalytic chemical vapour deposition route. Sodium dodecyl sulfate (SDS) ReagentPlus^®^ surfactant (≥98.5% purity) was supplied by SIGMA-ALDRICH Inc. (St. Louis, MO, USA). A set of clear glass vials of 40 mm outer diameter, 1.6 mm wall thickness, and 95 mm height were obtained from Glass Solutions Limited (Hertfordshire, UK) to host the liquid samples (i.e., water and nanofluids). A roll of sealing film that is made primarily of polyolefins and paraffin waxes was supplied by PARAFILM^®^ M (Neenah, WI, USA) and used to seal the open section of the glass vials after inserting the suspension content. The deionised water (DIW) that was used as the base fluid for synthesizing the nanofluids was produced by an Elga PR030BPM1-US Purelab Prima 30 water purification system (Elga LabWater company, Buckinghamshire, UK). Furthermore, the pH of the DIW was modified to 7 at an atmospheric temperature of 20 °C. This was done by adding sodium hydroxide (NaOH) solution (1.09956. Titrisol^®^), which was obtained from SIGMA-ALDRICH Inc. (St. Louis, MO, USA), to the liquid while being stirred and monitored using a calibrated HACH HQ11D portable pH meter (Loveland, CO, USA) that was connected to a PHC20101 Intellical gel filled Ph electrode (Loveland, CO, USA). The accuracy of the pH meter was reported by the manufacturer as ±0.002 pH, whereas the initial calibration of the pH device was conducted using commercial calibration fluids of pH 4, 7, and 10, which were obtained from Metrohm USA Inc. (Tampa, CA, USA). Figure 1 shows an illustration of the glass vial, nanopowder, SDS surfactant, and sealing film that were used in the experiment.

### 2.2. Nanopowder Characterization

An elemental test was performed for the MWCNT nanopowder through a 9 kW Rigaku SmartLab, Japan, X-ray diffraction (XRD) analyser (Rigaku Corporation, Tokyo, Japan) and its software, SmartLab Guidance (version 1.0), using a CuK_α_ X-ray source with a diffraction angle of 2θ and an incidence beam step of 0.1° to determine the Bragg’s peaks of the elements contained in the examined sample. The diffraction scanning angle range was from 10° to 80°, with a scanning rate of 1°/min. A JEOL JSM-IT700HR field emission scanning electron microscopy (FE-SEM) device (JEOL Ltd., Tokyo, Japan) and its integrated energy dispersive x-ray spectroscopy (EDS) analyser (JEOL Ltd., Tokyo, Japan) were used to check the morphology, MWCNTs outer diameters, and any impurities in with the as-received nanopowder feedstock. A small amount of the MWCNTs powder was placed on double-faced silver tape after attaching it on the device sample holder. Furthermore, the FE-SEM images were recorded at two different magnifications by the secondary electron mode from the surface region of the tested sample. Image-J software (version 1.53k, Wayne Rasband National Institutes of Health, Bethesda, MD, USA) was used to obtain the minimum, maximum, and average diameters of the MWCNT nanopowder from the FE-SEM high magnification image. The FE-SEM and EDS analyses were conducted at a working distance of 10 mm from the sample with an accelerating voltage of 20 kV to reduce any possible damage to the examined powder, and the operating software used was InTouchScope 1.12. The density of the MWCNT ($\rho_{\mathrm{MWCNT}}$) was obtained to calculate the required nanopowder vol %, which is part of the nanofluids fabrication procedure. This was done by first measuring the MWCNTs sample weight, using an ae-ADAM PW 214 analytical balance of 0.0001 g readability and ±0.0002 g accuracy. Then, the weighted powder was placed inside a HumiPyc Model 1 gas pycnometer–volumetric analyser (InstruQuest Inc., Boca Raton, FL, USA), which operated at 20 °C, to obtain the density of the sample from its input mass ($m_{\mathrm{MWCNT}}$) and the volume ($V_{\mathrm{MWCNT}}$) measured by the instrument. The following equation was used by the device to determine the powder sample density, which was found to be 2106 kg/m^3^.
(1)$$\rho_{\mathrm{MWCNT}}=\frac{m_{\mathrm{MWCNT}}\;}{V_{\mathrm{MWCNT}}}$$

### 2.3. Base Fluid Characterization

The density of the as-prepared water (i.e., adjusted to pH 7) was measured at temperatures ranging from 10 °C to 50 °C using a DMA 4500 M Anton–Paar GmbH company density meter device (Ashland, VA, USA), which has an auto calibration and cleaning feature. The device uses 20 mL of the tested sample to determine its property. The specific heat capacity of the water was measured at similar temperatures using a SETRAM Instrumentation LABSYS evo DTA/DSC device (KEP Technologies Inc., Austin, TX, USA). This was done by adding a small amount of the liquid, using a glass pipette dropper, in the device sample holder until it reached the height recommended by the manufacturer. Then the system operated at the specified temperature. In addition, the thermal conductivity of the water was then measured using a Thermtest company THW-L2 hot-wire apparatuses (Richibucto Road, NB, Canada). This was done by placing the liquid sample in the 100 mL clear glass beaker, which was placed on a hot/cold plate (supplied by Thermtest company, Richibucto Road, NB, Canada) that was set to the targeted temperature (i.e., 10 °C to 50 °C). Then the device probe was immersed into the test specimen. Three readings were obtained (5 min between each measurement), after which they were averaged to obtain the average thermal property value. The base-fluid viscosity was measured using a RV-2T viscometer (supplied by W&J Instrument LTD., Changzhou City, China) at a number of temperatures between 10 °C and 50 °C. Firstly, the device probe was immersed in the liquid (up to the reference line indicated on the probe) after being placed in the glass vial, then the scanning mode was used to determine a suitable rpm value. This value was used in the setting to obtain the as-produced water viscosity while controlling the sample temperature with the hot/cold plate.

### 2.4. Nanofluids Production

The chemical surfactant, SDS, of 1:1 (SDS:MWCNT weight ratio) was added to 80 mL of the as-adjusted water of pH 7, after which the added powder was dissolved using a Soniclean benchtop bath-type ultrasonic vibrator (Soniclean company, Dudley Park, South Australia, Australia) for 15 min while controlling the bath temperature from 10 °C to 50 °C. The selection of the surfactant ratio was based on a previous study that was conducted by Almanassra et al. [26], which showed that such a concentration of surfactant would provide a long-term physical stability (i.e., 6 months) to the suspension. Next, the nanofluid samples were prepared by placing the nanopowder first inside a new vial then injecting the as-prepared water–SDS surfactant, using a disposable syringe, on top of the nanopowder after which the vials were tightly sealed using the sealing film. The concentrations of nanopowders used were 0.01, 0.05, and 0.10 vol %, for each experimental set up. These were calculated through the mixing theory (Equation (2)) that is widely adopted by researchers working in this field [17].
(2)$$\mathrm{vol}.\;\%=\frac{V_{\mathrm{MWCNT}}\;}{V_{\mathrm{MWCNT}}+{\;V}_{\mathrm{bf}}}\times 100$$

The vials containing the solution were then placed gently in the Soniclean benchtop bath-type ultrasonic vibrator, running for 90 min at 100% power (43 kHz pulse) and filled with water to the operating level recommended by the manufacturer, to agitate the mixture at different temperatures (i.e., 10 °C–50 °C). This kind of particle dispersion method is known as the two-step approach, which is a common procedure used for the production of nanofluids by many researchers [27,28,29]. The bath temperature was maintained throughout the preparation phase by gradually adding cold/hot water inside the ultrasonic tank and extracting any excess water from the device via the attached ejection valve, and the surrounding temperature in which the experiments were conducted was 20 °C.

### 2.5. Nanofluid Effective Thermophysical Properties and Physical Stability

The effective density, specific heat capacity, thermal conductivity, and viscosity of the as-fabricated nanofluids were obtained using the same methods that were employed to obtain these properties for the base fluid. The properties were measured directly after the samples were produced. This was to minimize any changes in the physical stability of the dispersed particles. It is important to note that the temperature of the as-synthesized samples were maintained during the measurements of the properties using a dry bath solid block accessory, which was placed on a hot/cold plate device (Thermtest Co., Richibucto Road, NB, Canada). Moreover, the obtained thermophysical properties were later used in the mathematical modelling of the intercooler. In terms of the physical stability of the as-prepared dispersions, it was visually determined using the photographic image capturing method [18,30] directly after the preparation of the nanofluids and up to 45 days.

### 2.6. Intercooler System Modelling

The intercooler design that is considered in this study was adopted from the published work of Zhao et al. [14], and consists of a copper–nickel alloy plate–fin HE with a reverse flow configuration, as shown in Figure 2. The intercooler preliminary constructional data are presented in Table 1. In addition, the following assumptions were used in the analysis of the intercooler system:1-The fins used are straight on both liquid and gas sides of the HE.2-The liquid side (NL) has one fin layer fewer than the gas side (NG), i.e., NL = NG − 1.3-The intercooler operates at a steady-state condition.4-The fins thickness is uniform.5-The fins thermal resistance effect is negligible.6-The construction material of the intercooler parts is the same.7-The effect of corrosion and fouling build-up are neglected.

### 2.7. Theoretical Equations 

The intercooler system of the WR-21 is located between two compressor stages. For the first compressor stage, the inlet gas (i.e., air) temperature is 15 °C, after which the temperature is assumed to increase to 172.08 °C (first stage outlet temperature) due to the compression mechanism and as part of the design point. The compressed gas then gets introduced to the intercooler system, where it loses some of its thermal energy to the working fluid that is circulated within the system. The amount of heat transfer between the compressed gas and the coolant depends on several factors: the effectiveness of the intercooler (Ɛ), gas and coolant inlet temperatures, thermophysical properties of the gas and coolant, and both fluids’ flow rates. The flow rate of the gas and coolant were assumed to be 71.5 kg/s and 56.9 kg/s, respectively. The value of Ɛ, can be determined by using the following [14]:(3)$$Ɛ=\frac{1-\exp\left[-\mathrm{NTU}\left(1-\mathrm{Cr}\right)\right]}{1-\mathrm{Crexp}\left[-\mathrm{NTU}\left(1-\mathrm{Cr}\right)\right]}$$
where NTU is the number of transfer units, and Cr is the heat capacity ratio of the minimum heat capacity ($C_{\min}$) to the maximum heat capacity ($C_{\max}$). Hence, $\mathrm{Cr}=C_{\min}/C_{\max}$. To calculate NTU, the following Equations (4) and (5) can be employed.
(4)$$\frac{1\;}{\mathrm{NTU}}\;=\frac{C_{\min}}{\mathrm{UA}}$$
(5)$$\frac{1\;}{\mathrm{UA}}\;=\frac{1\;}{\eta_{\mathrm{ef},a}{\;h}_{a}{\;A}_{a}}+\frac{\delta_{\mathrm{pl}}\;}{{ƛ}_{\mathrm{pl}}{\;A}_{\mathrm{pl}}\;}+\frac{1\;}{\eta_{\mathrm{ef},b}{\;h}_{b}{\;A}_{b}}$$
where U, A, $\delta_{\mathrm{pl}}$, $\eta_{\mathrm{ef},a}$, $\eta_{\mathrm{ef},b},h_{a}$, $h_{b}$, $A_{a}$, $A_{b}$, $A_{\mathrm{pl}}$, and ${ƛ}_{\mathrm{pl}}$ are the overall heat transfer coefficient, heat transfer surface area, plate thickness, effective heat transfer surface efficiency at the gas side, effective heat transfer surface efficiency at the coolant side, convective heat transfer coefficient of the gas, convective heat transfer coefficient of the coolant, heat transfer surface area at the gas side, heat transfer surface area at the liquid side, plate heat transfer surface area, and thermal conductivity of the plate, respectively. 

The heat transfer coefficient of the gas turbulent flow was calculated by the equation that was used by Gnielinsk [31].
(6)$$h_{a}=\frac{N_{\mathrm{ua}}\;.{ƛ}_{a}}{D_{a}}$$
where ${ƛ}_{a}$, $N_{\mathrm{ua}},$ and $D_{a}$ are the thermal conductivity of the gas, Nusselt number, and hydraulic diameter, respectively. In order to determine the $N_{\mathrm{ua}}$ and $D_{a}$, the following Equations (7) and (8) were used.
(7)Nua = (fa2)(Rea−1000) Pra1+12.7(fa2)0.5(Pra 2/3−1)
(8)$$D_{a}=\frac{2\left(H_{a}-{\;t}_{f,a}\right)\left(S_{f,a}-{\;t}_{f,a}\right)\;}{H_{a}+S_{f,a}-2t_{f,a}}$$
where $f_{a}$, Rea, ${\mathrm{Pr}}_{a}$, $t_{f,a},{\;H}_{a}$, and $S_{f,a}$ are the Fanning factor, Reynolds number of the gas, Prandtl number of the gas, fin thickness, fin height, and fin pitch, respectively. Furthermore, the $f_{a}$, Rea, and ${\mathrm{Pr}}_{a}$ are defined as follow
(9)$$f_{a}\;=\;\frac{1}{{\left(1.58{\mathrm{lnRe}}_{a}-3.28\right)}^{2}}$$
(10)Rea = Wa DaAff,a µa 
(11)$${\mathrm{Pr}}_{a}=\frac{C_{p,a}{\;µ}_{a}\;}{{ƛ}_{a}}$$
where $W_{a}$, $A_{\mathrm{ff},a}$, ${µ}_{a},\;$and C_p,a_ are the mass flow rate of the gas, effective circulation area at the gas side, gas viscosity, and specific heat capacity of the gas, respectively. As for the heat transfer coefficient of the coolant, which is assumed to be in its laminar state, it can be calculated with respect to the Colburn factor ($j_{b}$) [32]:(12)$$h_{b}\;=\;\frac{j_{b}G_{b}C_{p,b}}{{\mathrm{Pr}}_{b}{}^{2/3}}$$
(13)$$j_{b}=\;\exp\left[0.103109{\left({\mathrm{lnRe}}_{b}\right)}^{2}-1.91091\left({\mathrm{lnRe}}_{b}\right)+3.211\right]\;$$
(14)$$G_{b}\;=\;\frac{W_{b}}{A_{\mathrm{ff},b}\;}$$
(15)Reb = Wb DbAff,b µb 
(16)$${\mathrm{Pr}}_{b}=\frac{C_{p,b}{\;µ}_{b}\;}{{ƛ}_{b}}$$
where $G_{b}$, Reb, ${\mathrm{Pr}}_{b}$, $W_{b}$, $D_{b},$ $A_{\mathrm{ff},b}$, ${µ}_{b},\;$and $C_{p,b}$ are the mass flow velocity of the coolant, Reynolds number of the coolant, Prandtl number of the coolant, mass flow rate of the coolant, coolant hydraulic diameter, effective circulation area of the coolant, coolant viscosity, and specific heat capacity of the coolant, respectively.

Both gas and coolant effective circulation areas (i.e., $A_{\mathrm{ff},a}$ and $A_{\mathrm{ff},b}$) were calculated using the following two equations:(17)$$A_{\mathrm{ff},a}\;=\;\frac{N_{a}\left(L2-2\delta_{s}\right)\;\left(H_{a}-t_{f,a}\right)\left(S_{f,a}-t_{f,a}\right)}{S_{f,a}}$$
(18)$$A_{\mathrm{ff},b}\;=\;\frac{N_{b}\left(L2-2\delta_{s}\right)\;\left(H_{b}-t_{f,b}\right)\left(S_{f,b}-t_{f,b}\right)}{S_{f,b}}$$

$A_{a}$, $A_{b}$, and $A_{\mathrm{pl}}$ of the intercooler were calculated with the below equations.
(19)$$A_{a}\;=\;2N_{a}L1\left(L2-2\delta_{s}\right)\left[1+\frac{\;\left(H_{a}-2t_{f,a}\right)}{S_{f,a}}\right]$$
(20)$$A_{b}\;=\;2N_{b}L1\left(L2-2\delta_{s}\right)\left[1+\frac{\;\left(H_{b}-2t_{f,b}\right)}{S_{f,b}}\right]$$
(21)$$A_{\mathrm{pl}}\;=\;L1L2\left[2+2\;\left(N_{a}-N_{b}\right)\right]$$

The heat transfer efficiency for both gas and coolant (i.e., $\eta_{\mathrm{ef},a}$ and $\eta_{\mathrm{ef},b}$) were calculated using the following.
(22)$$\eta_{\mathrm{ef},a}\;=\;\frac{\;\eta_{f,a}\left(H_{a}-t_{f,a}\right)+\left(S_{f,a}-t_{f,a}\right)}{S_{f,a}+H_{a}-2t_{f,a}}$$
(23)$$\eta_{\mathrm{ef},b}\;=\;\frac{\;\eta_{f,b}\left(H_{b}-t_{f,b}\right)+\left(S_{f,b}-t_{f,b}\right)}{S_{f,b}+H_{b}-2t_{f,b}}$$
where $\eta_{f,a}\;$and $\eta_{f,b}$ are the heat transfer efficiency of the fins at the gas and coolant side, respectively. The values of $\eta_{f,a}\;$and $\eta_{f,b}$ were determined as following.
(24)$$\eta_{f,a}\;=\;\frac{\;\tan\left(0.5m_{a}H_{a}\right)}{0.5m_{a}H_{a}}$$
(25)$$\eta_{f,b}\;=\;\frac{\;\tan\left(0.5m_{b}H_{b}\right)}{0.5m_{b}H_{b}}$$
(26)$$m_{a}=\sqrt{\frac{\;2h_{a}}{{ƛ}_{f,a}t_{f,a}}}$$
(27)$$m_{b}=\sqrt{\frac{\;2h_{b}}{{ƛ}_{f,b}t_{f,b}}}$$
where $m_{a}\;$and $m_{b}$ are the fins factors at the gas and coolant side, respectively. To obtain the heat transfer rate (Q), Equation (28) was used.
(28)$$Q=ƐC_{\min}\left(T_{\mathrm{in},a}-T_{\mathrm{in},b}\right)$$
where $T_{\mathrm{in},a}\;$and $T_{\mathrm{in},b}$ are the gas and coolant inlet temperatures, respectively. As for the coolant pressure drop ($P_{b}$), the following equation was used.
(29)$$P_{b}\;=\;\frac{\;2f_{b}L1G_{b}^{2}}{\rho_{b}D_{b}}$$
(30)fb = 24Reb(1−1.3553βb+1.9467βb2−1.7012βb3+0.9564βb4−0.2537βb5)
(31)$$\beta_{b}\;=\;\frac{{\;S}_{f,b}-{\;t}_{f,b}}{H_{b}-{\;t}_{f,b}}$$
where $f_{b}\;$and $\beta_{b}$ are the fanning factor and coolant channel aspect ratio, respectively. The pumping power (${\mathrm{Pump}}_{\mathrm{power}}$) was obtained using the $P_{b}$ and the volumetric flow rate of the coolant ($V_{b}$), as following.
(32)$${\mathrm{Pump}}_{\mathrm{power}}=P_{b}V_{b}$$
(33)$$V_{b}\;=\;\frac{{\;W}_{b}}{\rho_{b}}$$

### 2.8. Coolant and Gas Thermophysical Properties

The thermophysical properties of the coolant as well as the gas have a major effect on the intercooler performance, and therefore influence both the heat and mass transfer of these fluids when exiting the system. Since these properties are highly sensitive to temperature changes, they were obtained at different temperatures. For the coolant, these properties were measured from 10 °C to 50 °C, as was explained previously in Section 2.3 and Section 2.5; whereas for the gas, they were calculated using the formulas of Yang and Tao [33] that are shown in the Table 2.

For comparison purposes, the data on copper (Cu)–water nanofluids, which were obtained from the work of Zhao et al. [14], were also included in the study. It is important to note that Zhao et al. [14] have used 5 vol % of Cu nanoparticles that had a thermal conductivity, specific heat capacity, and density of 401 W/m∙K, 386 J/kg∙K, and 8960 kg/m^3^, respectively. In addition, the authors used a different correlation to determine the thermophysical properties of their nanofluids. These formulas can be found in their published work [14].

## 3. Results and Discussion

### 3.1. X-ray Diffraction Analysis

In the XRD analysis performed on the as-received MWCNT powder, the electromagnetic beam that is emitted from the X-ray source is reflected by the crystalline plane. The resulting reflection angle corresponds to the crystalline structure of the examined material. Thus, comparing the beam reflection results from the different crystalline planes to known reference values, the material’s spectra are revealed for the as-tested sample. By comparing the diffraction pattern obtained from the performed analysis (Figure 3) with other published works, such as Palanisamy and Kumar [34] and Sandhu and Gangacharyulu [35], it was noticed that similar results were acquired, therefore, the as-received powder is MWCNTs.

The high (0 0 2) peak that is seen in the XRD pattern is mostly due to the interlayer stacking of graphene sheets that are nested together, indicating the concentric cylindrical nature of these graphene sheets into forming MWCNTs [36]. It is important to note that the XRD pattern of graphite is relatively similar to that of MWCNTs due to the intrinsic nature of both materials [37]. The crystallite size at the highest peak was found through the Scherrer formula [38,39,40,41] to be 132.80 Å.
(38)$$D_{hkl}=\frac{F\lambda }{\beta_{hkl}cos\theta_{hkl}}$$
where F represents the shape factor that has a constant value of ~0.9, λ signifies the wavelength of the CuK_α_ X-ray radiation source employed and is equal to 0.15405 nm, $\beta_{hkl}$ is the full width at half the maximum of the (hkl) diffraction peak, and $\theta_{hkl}$ is the Bragg angle at the (hkl) peak. 

### 3.2. FE-SEM and EDS Characterization

The FE-SEM analysis of the as-received nanopowder showed the existence of some agglomerations between the particles, which can be linked to their high surface energy which results from the high surface-to-volume ratio of the MWCNTs. Thus, the particles tend to attach to each other in order to reduce their surface energy and reach a more stable thermodynamic state. The aforementioned observation can be clearly seen through the SEM patterns in Figure 4a,b. In terms of morphology, the outer diameter of the MWCNTs was shown to be roughly in the range of 26 to 132 nm, with the median value and standard deviation (S.D) being 54.87 nm and 18.86, respectively (Figure 4c). Comparing the range of diameters as-reported by the manufacturer (i.e., 30 to 50 nm) to those obtained from the performed FE-SEM analysis and Image-J software (Figure 4c), demonstrated that there are some variations in the supplied values of up to 82 nm. Such variation in diameter in commercial MWCNT powders is normal and was previously experienced by Singh et al. [42]. It can be caused by variations in the manufacturing process and/or the oxidation of the sample from the unavoidable exposure of the sample to the surrounding atmosphere before undergoing the characterization stage. Furthermore, the EDS elemental mapping (Figure 4d–h) has shown the presence of nickel (Ni) and copper (Cu) elements within the as-characterized MWCNTs sample, which is due to the Ni coating of the sample and Cu adhesive tape placed under the powder sample.

### 3.3. Thermophysical Properties Measurements and Physical Stability

In this section, the thermophysical properties of the base fluid and as-produced nanofluids are demonstrated as well as the physical stability of the suspensions. This includes the density, specific heat capacity, thermal conductivity, and viscosity, where their measured data can be seen in Table 3. For the density, two observations were noticed. The first is that, increasing the dispersed MWCNTs concentration causes the property to increase, however increasing the temperature of the fluid results in reducing its density. In addition to the previous observation, it was also noticed that the density of the nanofluid samples were very close to that of the base fluid. This can be attributed to the low concentration of nanomaterial used in their preparation. Furthermore, the highest and lowest values recorded for the base fluid were 0.9997 g/cm^3^ (at 10 °C) and 0.9880 g/cm^3^ (at 50 °C); whereas for the 0.1 vol % sample, it was shown to be 1.0008 g/cm^3^ (at 10 °C) and 0.9920 g/cm^3^ (at 50 °C), respectively. When comparing the obtained density values of the base fluid (i.e., water) with those available in the literature (e.g., Baboian [43] published work), it showed an average deviation of about 1.18%, and hence the measurements can be considered reliable due to the low divergence between the results and published data. As for the specific heat capacity, the changes in the property due to temperature variation have not been shown to follow a certain trend, however the property proved to be very sensitive to the vol % of the dispersed nanomaterial. For instance, the base fluid showed values between 4178 J/kg∙K (lowest) and 4193 J/kg∙K (highest) at 30 °C and 10 °C, respectively. However, the 0.1 vol % samples had the lowest specific heat capacity values, which were 3181 J/kg∙K (lowest) and 3205 J/kg∙K (highest) at 10 °C and 50 °C. Such property behaviour is common with suspensions of a similar nature [44], and can be explained by the high thermal conductivity of the dispersed nanomaterials that causes the mixture effective thermal transportation to increase but at the same time reduces its thermal storage capability (i.e., specific heat capacity). As for the thermal conductivity, the enhancement in the property is proportional to the increase in the temperature of the fluid and the concentration of dispersed MWCNTs. For the base fluid, raising the temperature from 10 °C to 50 °C caused the thermal property to increase from 0.582 W/m∙K to 0.641 W/m∙K, which corresponds to an enhancement of ~10.14%. On the other hand, the thermal conductivity of the 0.1 vol % nanofluids showed values of 0.609 W/m∙K (at 10 °C) and 1.075 W/m∙K (at 50 °C), which correspond to ~76.52% enhancement in the property from raising the suspension temperature alone. The previous results clearly show the superiority of MWCNTs-based nanofluids over that of the base fluid as well as the sensitivity of the property towards the nanomaterial loading and fabrication temperature. Nevertheless, it must be mentioned that the added surfactant has a negative effect on the thermal conductivity of the mixture (i.e., it reduces the thermal property) [17]. This is because when adding the SDS surfactant, a thin layer forms on the outer surface of the nanomaterial in order to modify the attraction/repulsion force between the dispersed particles and the surrounding environment, and hence minimum nanomaterial clustering occurs (i.e., better physical stability can be achieved). However, this newly introduced layer not only has a lower thermal conductivity than the MWCNTs but also acts as a resistance bridge between the direct interaction of the carbon-based nanomaterial and the base fluid, and therefore reduces the influence of the thermal conductivity of the dispersed particles on the host liquid. Furthermore, the results from the effective viscosity measurements of the samples showed that this property is significantly affected by changes in temperature, whereas the particle concentration had low influence on the viscosity due to the low quantity of MWCNTs used in loading. For example, raising the base fluid temperature from 10 °C to 50 °C showed a decrease in the viscosity from 1.280 mPa∙s to 0.525 mPa∙s, which corresponds to ~59.38% reduction. Moreover, the viscosity of the 0.1 vol % nanofluids showed to be 1.344 mPa∙s and 0.534 mPa∙s at 10 °C and 50 °C, respectively. These values correspond to a reduction of 41.02% from increasing the suspension temperature (i.e., from 10 °C to 50 °C). When comparing the level of increase in viscosity between the base fluid and the high concentration samples (i.e., 0.1 vol % nanofluids), the results showed that the suspensions are ~4.7 % and ~0.04 % higher than that of the base fluid at 10 °C and 50 °C, respectively. Such low level of increase in the viscosity is considered favourable in heat transfer applications because it reduces the pumping power requirements [1].

In addition to the previous properties, an illustration of the stability behaviour of suspensions fabricated at 30 °C is shown in Figure 5a–d. It was found that the as-prepared samples showed good stability up to 10 days (Figure 5b). However, the low MWCNTs concentration samples started to lose a lot of its stability after day 10 (Figure 5c,d). On the other hand, the 0.05 vol % and 0.10 vol % samples remained stable even on day 45 (Figure 5d). This suggest that the concentration of SDS used in forming the 0.01 vol % was not sufficient to change the head group charge of the dispersed MWCNTs due to the low volume ratio of surfactant to base fluid used [17].

### 3.4. Intercooler Modelling Output 

The modelling results from operating the intercooler system with different coolants, i.e., water, MWCNTs-based nanofluids, and Cu-based nanofluid are presented in the following subsections.

#### 3.4.1. Intercooler Effectiveness

In Figure 6, it can be seen that the effectiveness of the HE system improves when adding nanoparticles to the base fluid and/or raising the coolant inlet temperature. Therefore, the level of enhancement at a fixed gas inlet temperature can be seen to be dependent mainly on three elements, which are the type of nanomaterial used, concentration of the dispersed nanomaterial, and the inlet temperature of the coolant. As such, both types of nanofluids surpass the conventional working fluid (i.e., water). For example, the effectiveness of the intercooler with water, 0.1 vol % MWCNTs-based nanofluid, and 5.0 vol % Cu-based nanofluid have been shown to be 81.41%, 81.59%, and 82.29%, respectively, at 10 °C; whereas at 50 °C, the values were 82.35%, 84.06%, and 82.82%, respectively. However, when further inspecting the data in Figure 6, it can be observed that when the coolant temperature is below 30 °C the enhancement of the intercooler’s effectiveness was much higher with the Cu-based nanofluid than that of the MWCNTs-based nanofluids. Increasing the MWCNTs concentration to 0.05 vol. % and 0.10 vol. % as well as raising these MWCNTs-based coolants inlet temperature (i.e., ≥30 °C) overcomes the level of enhancement that is gained from the Cu-based nanofluid by ~1.25%, as was determined with the 0.10 vol. % MWCNTs-based nanofluid. Although this level of enhancement is small, nevertheless the volume of nanoparticles used to fabricate the 0.10 vol % MWCNTs-based nanofluid is 49 times fewer than that of the Cu-based working fluid.

#### 3.4.2. Intercooler Heat Transfer Rate

The heat transfer rate was seen to increase with the increase in the nanomaterial concentration and reduce with the increase in the coolant inlet temperature. Figure 7 illustrates the changes in the heat transfer rate with respect to the coolant inlet temperature. The heat transfer rate values at 10 °C were 11,745.78 kW, 11,771.36 kW, and 11,872.20 kW; and at 50 °C were 9468.82 kW, 9665.34 kW, and 9523.04 kW when using water, 0.10 vol. MWCNTs-based nanofluid, and Cu-based nanofluid, respectively. A similar observation can be made regarding the effect of the coolant inlet temperature below and above 20 °C between the Cu-based suspension and the MWCNTs-based coolants that are higher than 0.01 vol. %.

#### 3.4.3. Intercooler Gas and Coolant Outlet Temperatures 

As for both gas and coolant outlet temperatures, Figure 8, Figure 9 and Figure 10 demonstrate the changes caused to these two fluids. In general, the different suspensions have shown a maximum decrease of ~2 °C in the outlet gas temperature. To be more specific, at 10 °C, the exiting gas temperature was found to be 46.62 °C, 46.27 °C, and 44.89 °C, whereas at 50 °C the corresponding values were 77.71 °C, 75.02 °C, and 76.97 °C for the water, 0.10 vol. MWCNTs-based nanofluid, and Cu-based nanofluid, respectively. These results lead to a number of conclusions, which are as follows:a-The MWCNTs concentration used was not sufficient to obtain substantial improvements in the HE performance.b-The type and material of the HE used as an intercooler should be the subject of an optimization study.c-The flow rate used was high, and thus not best suited for providing sufficient heat transfer mechanism between the two fluids (i.e., gas and coolant).d-The intercooler inlet gas temperature was too low for the suspension to cause significant reduction in its exiting temperature.

To further justify the inlet gas temperature conclusion (i.e., the previously mentioned point d), the temperature of the gas entering the intercooler was increased from 172.08 °C to 407 °C. Figure 9 shows the new changes in the gas outlet temperature with the different coolants. The results show that the new level of enhancement can reach up to ~6.46 °C using the 0.10 vol % carbon-based nanofluid with 50 °C coolant inlet temperature, and ~3.68 °C using the 5.0 vol % Cu-based suspension with 10 °C coolant inlet temperature. As for the coolant outlet temperature (Figure 10), adding nanomaterial to the base fluid (i.e., water) resulted in increasing the coolant exiting temperature. This is due to the fact that these nanomaterials exhibit higher thermal conductivity than the base fluid itself, and hence would result in increasing the effective thermal conductivity of the suspensions and correspondingly the working fluid outlet temperature. Nevertheless, the highest increase in the coolant outlet temperature over that of the water was shown to be 3.71% for the 0.10 vol. % MWCNTs-based nanofluid (at 10 °C), whereas the lowest was 0.44% for the 0.01 vol. % MWCNTs-based nanofluid (at 40 °C). However, the 3.71% increase in coolant temperature in the HE is still considered low and insufficient to cause a significant impact, as was explained previously (i.e., Figure 8).

#### 3.4.4. System Pumping Power Requirement 

The pumping power requirement was found to increase with the increase in nanomaterial concentration and decrease with the rise in coolant temperature, as illustrated in Figure 11. For example, at 10 °C and 50 °C, using pure water as coolant was found to require a pumping power of 66.19 W (10 °C) and 31.05 W (50 °C), whereas the 0.10 vol % MWCNTs- and 5.0 vol % Cu-based coolants showed 79.54 W and 79.64 W at 10 °C, and 33.51 W and 37.93 W at 50 °C, respectively. The reason behind this behaviour is that increasing the concentration of the dispersed nanomaterial causes an increase in the shear stress of the suspension, and therefore a higher pumping power would be needed to drive the coolant. On the other hand, increasing the coolant temperature would lower its viscosity, and hence would consequently decrease the required pumping power. Since the thermophysical properties and concentration of the dispersed MWCNTs and Cu react differently to temperature changes, increasing the coolant inlet temperature would have different effects on the pumping power between those two nanomaterials, as is clearly seen in Figure 11. In general, it is commonly expected that a nanomaterial of higher thermal conductivity and lower concentration would require less pumping power when the suspension temperature is raised, and vice versa. The previous assumption is usually valid if the density of the dispersed nanomaterials is not large enough to have a major impact on the liquid effective viscosity.

Despite the previous outcomes, which indicate that the level of enhancement caused by the nanofluid in this particular study is low, given the low MWCNTS concentrations investigated, it is still believed that these results can be considered fairly promising if similar coolants are adopted for future designs of marine gas turbines with high gas temperatures and power outputs. It can also be seen as a starting point for adopting nanofluids in intercoolers of different geometrical designs.

## 4. Conclusions

In the present study, the performance of a marine gas turbine intercooler that utilizes MWCNTs-water nanofluids was investigated. The carbon-based nanofluids were fabricated using a two-step controlled-temperature method at a temperature range of 10 °C to 50 °C. The concentration of the MWCNTs suspensions were of 0.01–0.10 vol % and the surfactant used was SDS with a SDS to MWCNTs weight ratio of 1:1. Initially, the thermophysical properties of the nanomaterial and base fluid were obtained using XRD, FE-SEM, EDS, hot-wire, viscometer, pycnometer–volumetric analyser, and DTA/DSC devices. After the nanofluids preparation phase, their thermophysical properties were determined. This included the density, thermal conductivity, viscosity, and specific heat capacity. It was found that with higher preparation temperature, the density and viscosity of the suspension reduced. However, the thermal conductivity of the nanofluids reacted in the opposite manner with the rise in production temperature. On the other hand, the specific heat capacity did not appear to follow a specific trend with the changes in nanofluid temperature. As for the nanomaterial concentration, it was found that adding more MWCNTs to the base fluid has caused the density, viscosity, and thermal conductivity to increase, and the specific heat capacity to decrease. Furthermore, the as-prepared nanofluids as well as the one adopted from the literature (i.e., the Cu-based nanofluid) all improved the performance of the intercooler. However, the actual level of enhancement proved to be low, given the low concentration of MWCNTs employed in the study. When increasing the temperature of the gas entering the intercooler, the improvement in the intercooler performance showed an increase. Moreover, the MWCNTs have demonstrated a much better level of performance enhancement, notwithstanding a 49 times lower concentration when compared to that of the Cu-based nanofluid. The general conclusion that can be drawn from this research is that nanofluids tend to have a stronger effect when their exposure duration is higher in the intercooler system, and therefore the design of the HE is just as important as the working fluid. In addition, having a high gas inlet temperature would increase the effectiveness of the suspension.

## Figures and Tables

**Figure 1 nanomaterials-11-02300-f001:**
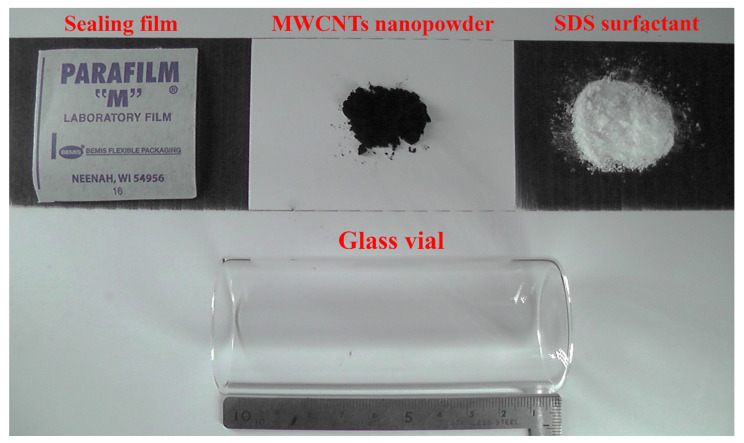
Nanofluids starting materials, which includes the sealing film, MWCNTs nanopowder, SDS surfactant, and glass vial.

**Figure 2 nanomaterials-11-02300-f002:**
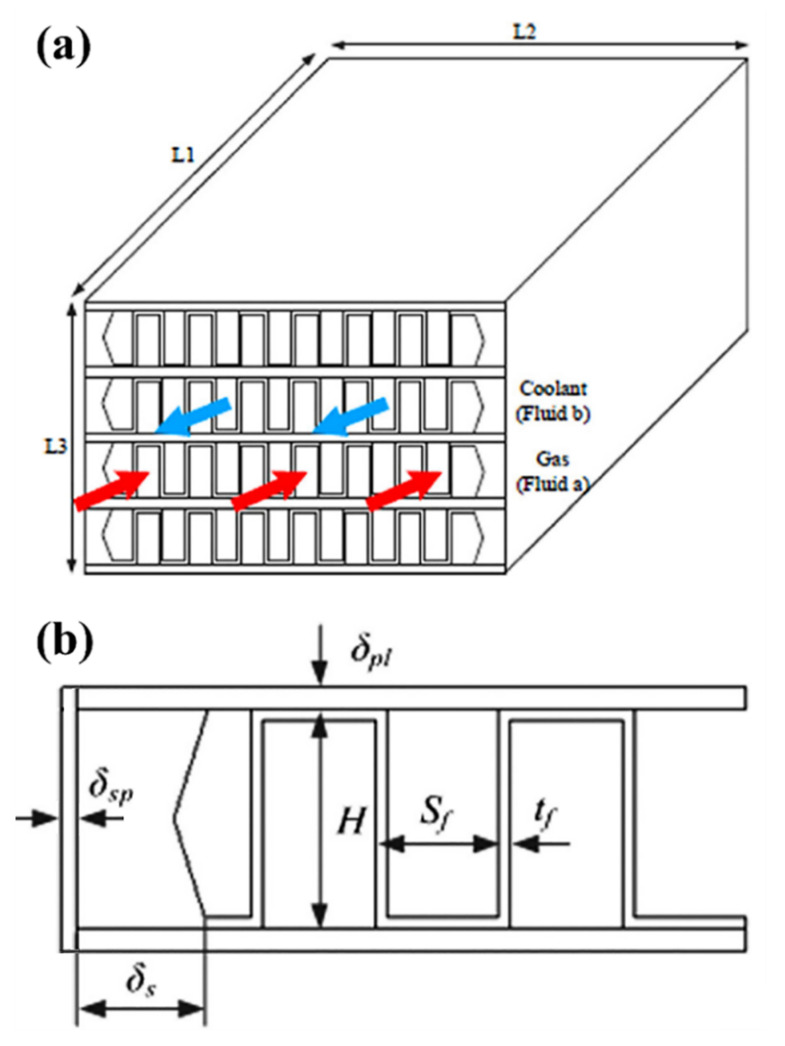
Illustration of the modelled intercooling system, where (**a**) shows the schematic arrangement of the HE, and (**b**) demonstrates the details of the straight fins.

**Figure 3 nanomaterials-11-02300-f003:**
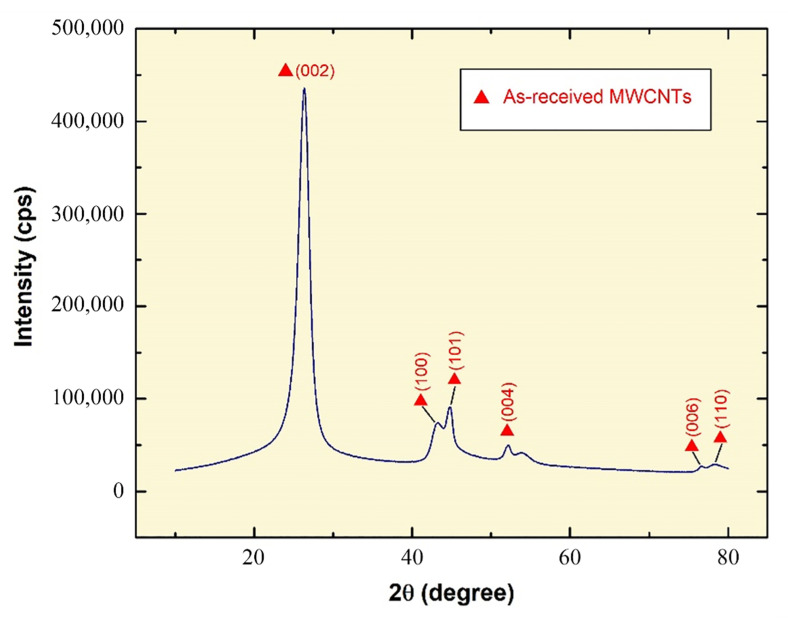
X-ray diffraction patterns of as-received MWCNTs powder.

**Figure 4 nanomaterials-11-02300-f004:**
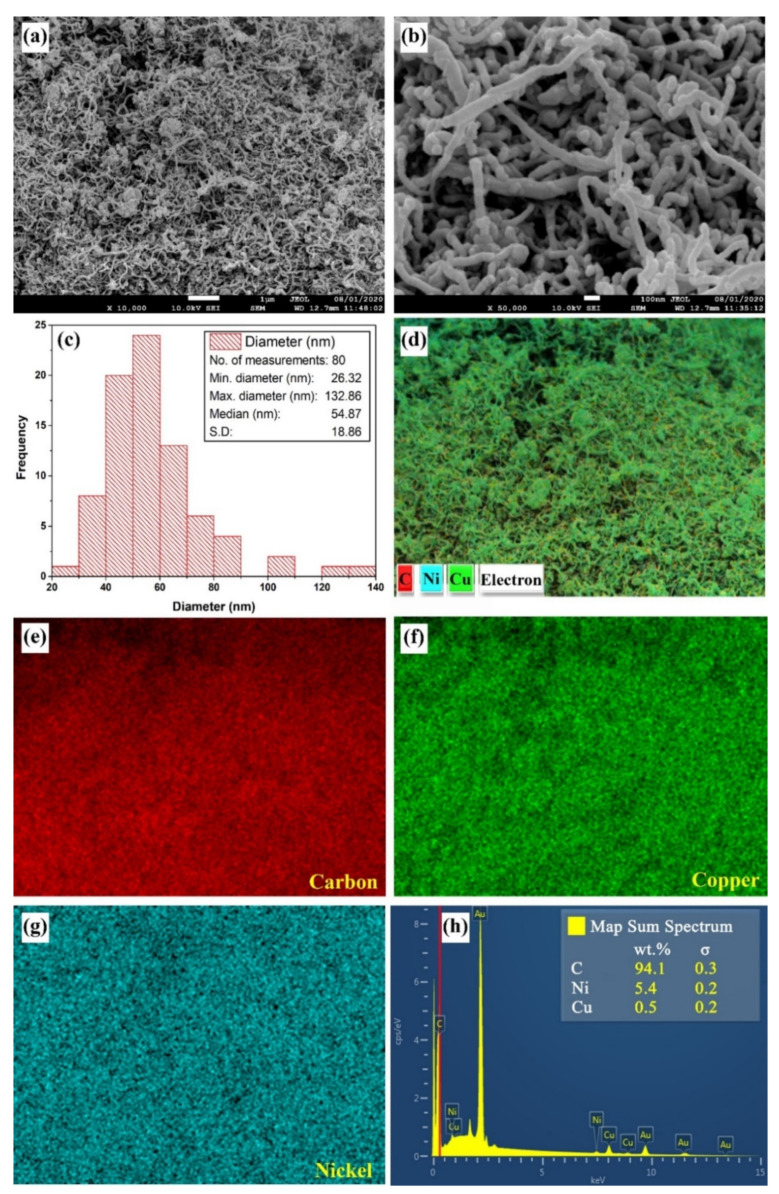
FE-SEM and EDS characterization, where (**a**–**b**) shows the particles morphology at low and high magnifications, (**c**) illustrates the diameter distribution, (**d**–**g**) demonstrates the elemental mapping, and (**h**) shows the x-ray spectrum.

**Figure 5 nanomaterials-11-02300-f005:**
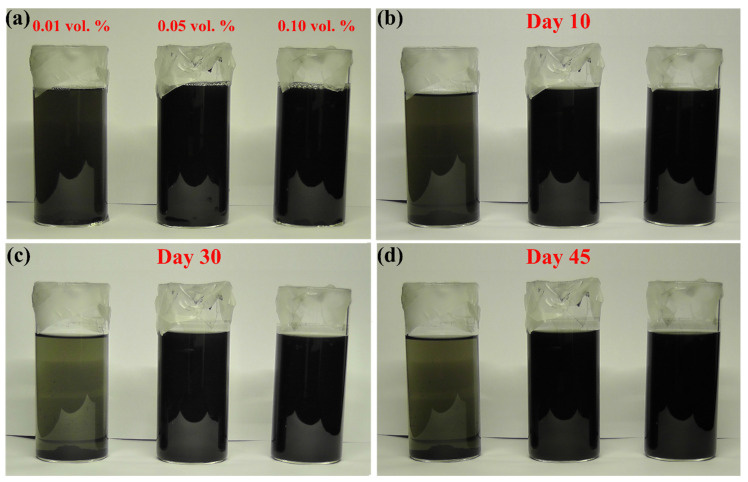
Image of the 30 °C controlled temperature as-prepared MWCNTs-based nanofluids, where (**a**) are the samples directly after production, (**b**–**d**) on days 10, 30, and 45, respectively.

**Figure 6 nanomaterials-11-02300-f006:**
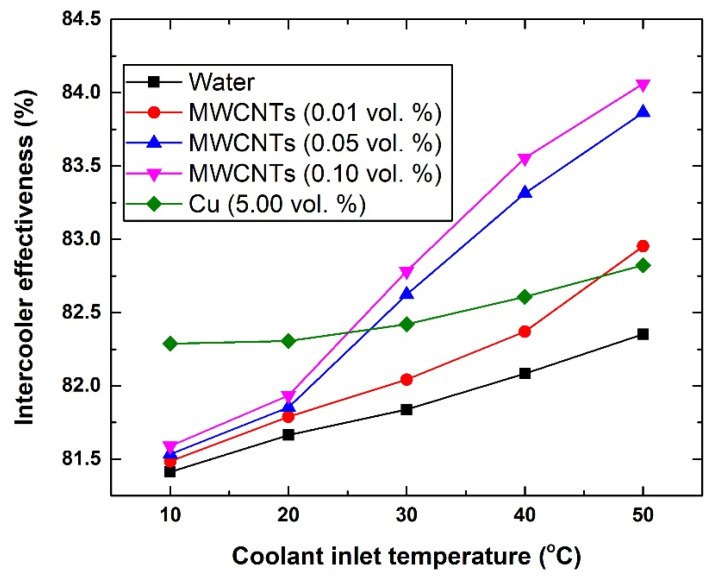
Changes in intercooler effectiveness with respect to coolants inlet temperature.

**Figure 7 nanomaterials-11-02300-f007:**
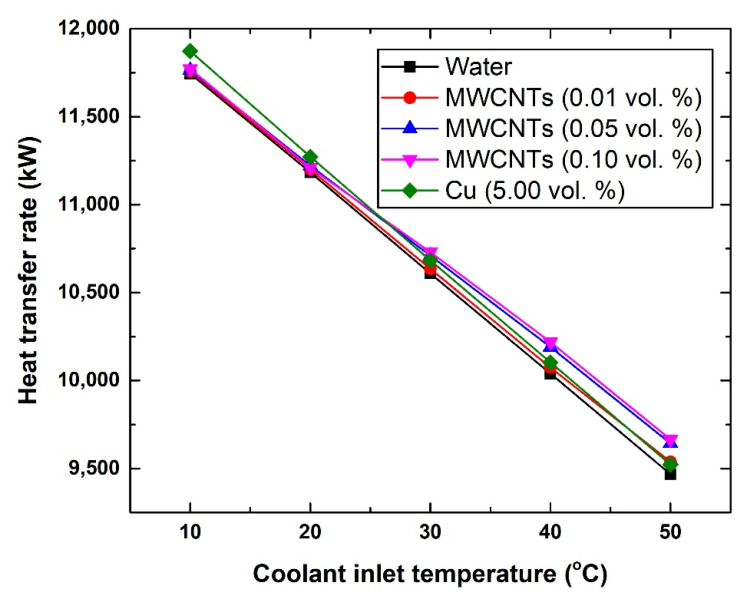
Changes in the heat transfer rate with respect to coolants inlet temperature.

**Figure 8 nanomaterials-11-02300-f008:**
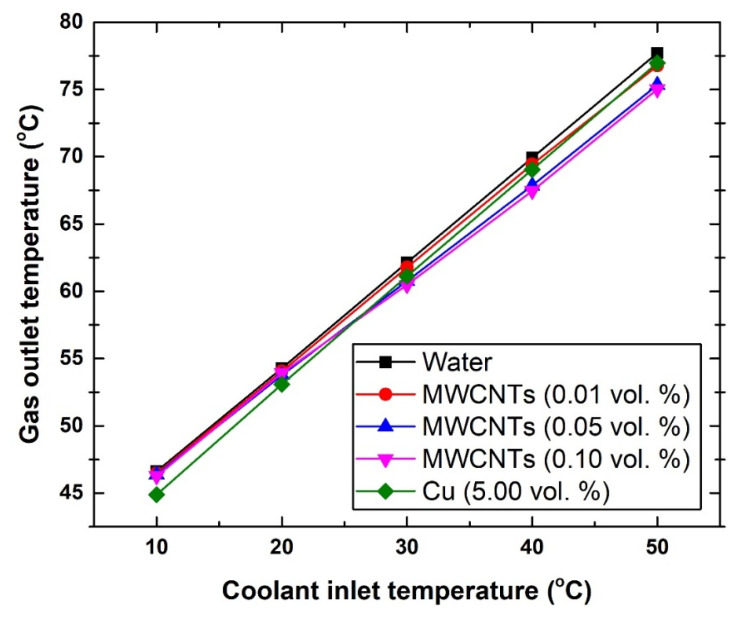
Changes in gas outlet temperature with respect to coolant inlet temperature.

**Figure 9 nanomaterials-11-02300-f009:**
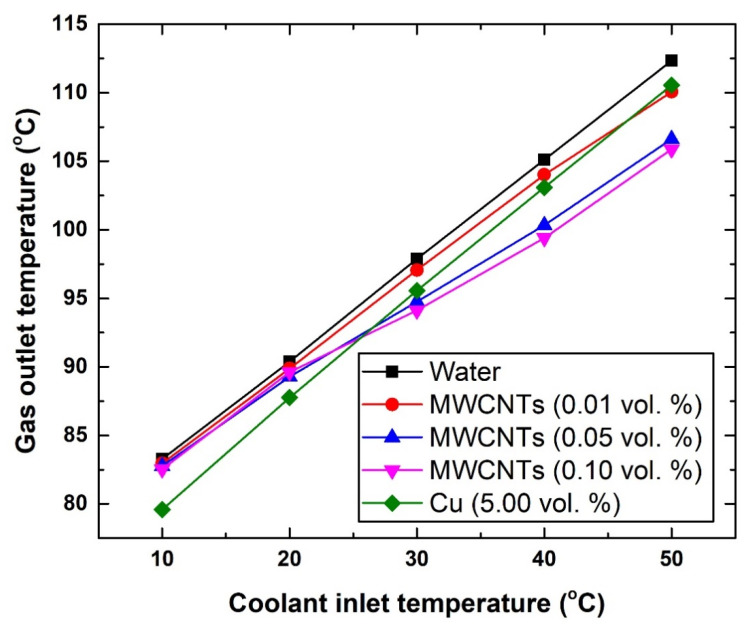
Changes in gas outlet temperature with respect to coolant inlet temperature and a fixed gas inlet temperature of 407 °C.

**Figure 10 nanomaterials-11-02300-f010:**
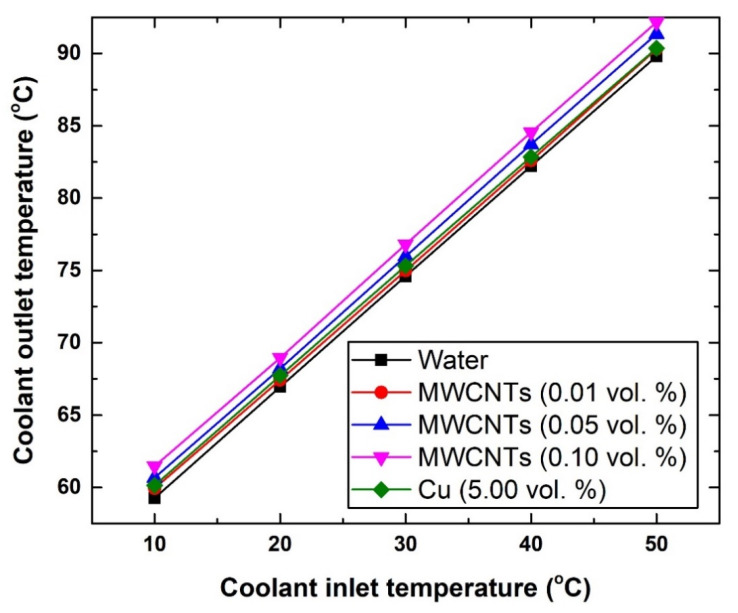
Changes in coolant outlet temperature with respect to coolant inlet temperature.

**Figure 11 nanomaterials-11-02300-f011:**
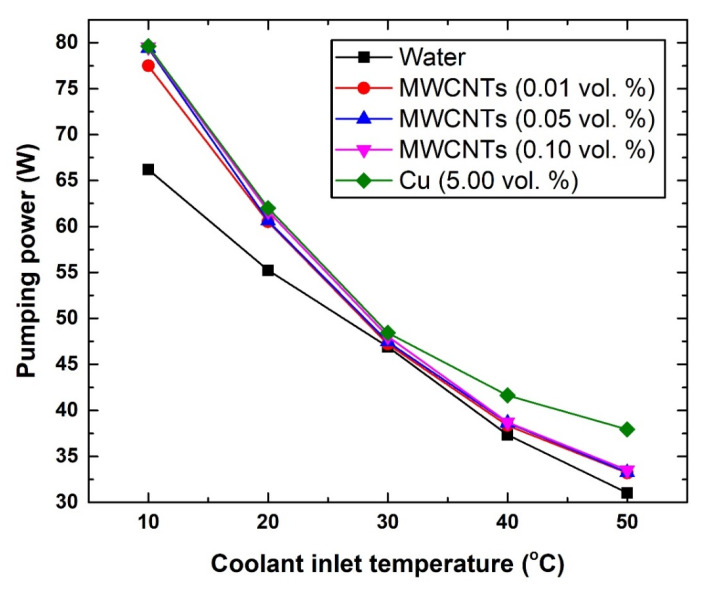
Changes in pumping power with respect to coolant inlet temperature.

**Table 1 nanomaterials-11-02300-t001:** Constructional data of the modelled intercooler.

Parameter	Value
Intercooler length (L1)	0.35 m
Intercooler width (L2)	1.0482 m
Intercooler thickness (L3)	0.0209 m
Plate thickness ($\delta_{\mathrm{pl}}$)	5.0 × 10^−4^ m
Side plate thickness ($\delta_{\mathrm{sp}}$)	1.6 × 10^−3^ m
Seal thickness ($\delta_{s}$)	6.0 × 10^−3^ m
Number of fin layers at the gas side ($N_{a}$)	36
Number of fin layers at the coolant side ($N_{b}$)	35
Fin thickness at the gas side ($t_{f,a}$)	2.0 × 10^−4^ m
Fin thickness at the coolant side ($t_{f,b}$)	2.0 × 10^−4^ m
Fin height at the gas side (H_a_)	6.2 × 10^−3^ m
Fin height at the coolant side ($H_{b}$)	3.0 × 10^−3^ m
Fin pitch at the gas side ($S_{f,a}$)	1.4 × 10^−3^ m
Fin pitch at the coolant side ($S_{f,b}$)	1.4 × 10^−3^ m

**Table 2 nanomaterials-11-02300-t002:** Formulas for calculating the thermophysical properties of the gas.

Property	Formula	Equation Number
Density ($\rho_{a}$)	$\rho_{a}=-1.487\times 10^{-9}T_{\mathrm{in},a}^{3}+3.638\times 10^{-6}T_{\mathrm{in},a}^{2}-0.003088T_{\mathrm{in},a}+1.2435$	(34)
Specific heat capacity ($C_{p,a}$)	$C_{p,a}=4\times 10^{-4}T_{\mathrm{in},a}^{2}-0.02T_{\mathrm{in},a}+1003$	(35)
Thermal conductivity (${ƛ}_{a}$)	${ƛ}_{a}=2.456\times 10^{-4}T_{\mathrm{in},a}^{0.823}$	(36)
Viscosity (${µ}_{a}$)	${µ}_{a}=1.50619\times 10^{-6}\left(\frac{T_{\mathrm{in},a}^{1.5}}{T_{\mathrm{in},a}+122}\right)T_{\mathrm{in},a}^{0.823}$	(37)

**Table 3 nanomaterials-11-02300-t003:** Measured thermophysical properties of the base fluid and as-prepared nanofluids.

Nanomaterial Concentration (vol %)	Nanofluid Temperature (°C)	Density (g/cm^3^)	Specific Heat Capacity (J/kg∙K)	Thermal Conductivity (W/m∙K)	Viscosity (mPa∙s)
Zero	10	0.9997	4193	0.582	1.280
	20	0.9981	4183	0.601	0.962
	30	0.9956	4178	0.614	0.783
	40	0.9921	4180	0.629	0.635
	50	0.9880	4181	0.641	0.525
0.01	10	0.9999	4061	0.592	1.305
	20	0.9984	4073	0.624	0.979
	30	0.9959	4079	0.651	0.792
	40	0.9924	4085	0.681	0.638
	50	0.9884	4079	0.751	0.527
0.05	10	1.0002	3639	0.601	1.339
	20	0.9988	3651	0.651	1.001
	30	0.9966	3657	0.767	0.799
	40	0.9934	3663	0.871	0.642
	50	0.9897	3669	0.983	0.530
0.10	10	1.0008	3181	0.609	1.344
	20	0.9993	3193	0.674	1.007
	30	0.9974	3199	0.813	0.809
	40	0.9950	3199	0.957	0.650
	50	0.9920	3205	1.075	0.534

## Data Availability

All data can be provided upon request from any of the authors.

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
