# Peer review of "Effect of Multi-Walled Carbon Nanotubes-Based Nanofluids on Marine Gas Turbine Intercooler Performance"

_nanomaterials, 2021, doi:10.3390/nano11092300_

Round 1

Reviewer 1 Report

To author:

This paper investigates the effect of utilizing nanofluids for enhancing the performance of a marine gas turbine intercooler. I think this is a very practical and valuable research. However, the length of this paper is too long to make the topic difficult to highlight. The structure of the paper needs further improvement. Some recommendations need to clarify and revise as follows.

Reviewer comments:

  1. Judging from the format of the journal's final manuscript, it is too long for a research paper on a single topic to reach 28 pages. You have to review the paper and delete unnecessary figures and repetitive content to make the paper more concise.
  2. The last paragraph of the Introduction section can be further reduced. The content of this paragraph is too repetitive with the subsequent related descriptions.
  3. The relevant data of the fundamental characteristics measurement of nanomaterials and nanofluids (Figs. 6~9) can be summarized in the second section with a table.
  4. Your nanomaterials are commercial products (not self-made), so you only need to clearly explain the source of the materials and do some simple tests to confirm the important properties. This paper is too large and trivial in this section. Please reduce this part.
  5. The relevant performance analysis of the marine gas turbine intercooler can be independently divided into the third section.
  6. The Results and Discussion section should focus on the analysis results of nanofluids applied to intercoolers. Please also refer to the third point above.
  7. You should state the stability of the nanofluid (the time of stable suspension). The stability of the nanofluid strongly affects the reliability of your intercooler performance analysis results.
  8. The format of the references must be revised in accordance with the journal's regulations.

Author Response

The authors are very much thankful to the respected reviewer for accepting to review our manuscript and sharing his valuable time. We cordially acknowledge the useful comments and recommendations made by the respected reviewer on our manuscript. We have tried to revise the manuscript accordingly and the detailed corrections are listed below point by point. The respected reviewer can find all the modifications and corrections in the revised manuscript written in red text. The authors respond to the respected reviewer queries are as follows:

Reviewer comments:

1. Judging from the format of the journal's final manuscript, it is too long for a research paper on a single topic to reach 28 pages. You have to review the paper and delete unnecessary figures and repetitive content to make the paper more concise.

  • We thank the respected reviewer for his respected suggestion, which we have done our best to do. Kindly note that we tried to provide the readers with as much information as possible on what we have done in this research, especially since MDPI Nanomaterials have no page or word limits for their articles. Nevertheless, we have tried to shorten the manuscript as much as possible based on the respected reviewer suggestion and request. Thank you very much sir.

2. The last paragraph of the Introduction section can be further reduced. The content of this paragraph is too repetitive with the subsequent related descriptions.

  • We thank the respected reviewer for his respected suggestion. Kindly note that we have reduced the text contain, as suggested. Please see the last paragraph in the Introduction Section. Thank you very much sir.

3. The relevant data of the fundamental characteristics measurement of nanomaterials and nanofluids (Figs. 6~9) can be summarized in the second section with a table.

  • We thank the respected reviewer for his respected suggestion. We are assuming that the respected reviewer would like us to remove Figs. 6-9 (old manuscript version) and replace them with a single Table to reduce the size of the manuscript, and therefore we have done that and corrected the corresponding text. If what we understood is wrong, we apologies and please let us know and we will be happy to include the removed Figures. Please see Table 3. Thank you very much.

4. Your nanomaterials are commercial products (not self-made), so you only need to clearly explain the source of the materials and do some simple tests to confirm the important properties. This paper is too large and trivial in this section. Please reduce this part.

  • We thank the respected reviewer for his respected comment and suggestion. Kindly note that we have the Table, which was in Section 3.1 and reduced the text in this section. We have also reduced the text in Section 3.2 as much as possible. Kindly note that we had to keep part of the analysis because their maybe some manufacturing defects with the nanopowder, and therefore had to do the analysis to show the reader the condition of our feedstock. However, kindly note that we were very happy to take your suggestion into consideration and have done the modifications in the revised version of the manuscript. Thank you very much sir.

5. The relevant performance analysis of the marine gas turbine intercooler can be independently divided into the third section.

  • We thank the respected reviewer for his respected suggestion, which we have taken into account in our revised manuscript. Kindly note that we have divided Section 3.4 into subsections 3.4.1 to 3.4.4, as requested. Thank you very much sir.

6. The Results and Discussion section should focus on the analysis results of nanofluids applied to intercoolers. Please also refer to the third point above.

  • We thank the respected reviewer for his respected comment. Kindly note that the authors have revised the work and focused more on the nanofluids as requested. As for point 3, the authors would like to keep the figures please and not replace them with a single table because in such case most of our work will be presented as values only. We do apologies for not fulfilling this part of the respected reviewer request. Please note that we may be confused on the actual reviewer request as to just adding a Table and keeping the figures or removing the figures and keeping the Table only. Kindly accept our sincere apology. Thank you very much.

7. You should state the stability of the nanofluid (the time of stable suspension). The stability of the nanofluid strongly affects the reliability of your intercooler performance analysis results.

  • We thank the respected reviewer for his suggestion, which we were happy to take into account. Kindly note that the stability of the nanofluids have been included in the revised version of the manuscript. Please check the red text in Section 2.5, Section 3.3, and Fig. 5. Thank you very much.

8. The format of the references must be revised in accordance with the journal's regulations.

  • We thank the respected reviewer for pointing this out. Kindly note that we have used Endnote software to code the references in the manuscript so that the journal can find it easy to change the current format to theirs with a single click. Thank you very much sir.

Finally, the authors would like to thank the respected reviewer for his time and very useful comments and remarks. Thank you very much.

Reviewer 2 Report

This paper presents an interesting contribution on the use of multi-walled carbon nanotubes-based nanofluids as coolants, applied in this case in a marine gas turbine intercooler.

In my opinion, before being published, the authors should correct or improve some aspects:

1) There is some confusion about the as-synthesized samples at different temperatures and the measurement of their properties at those temperatures. For instance, is the nanofluid as-prepared at 50ºC always kept at that temperature, to measure its properties? The authors should explain this point more clearly.

2) Line 131, page 3: "A set of clear glass vials of 40 mm outer diameter, 1.6 mm wall thickness, and 95 mm height were obtained..." However, in Figure 1 we can note that the height of the glass vial is greater than 95 mm. In the photograph of Fig. 5, the height of the glass vials is correct. It seems that the vial that appears in Figure 1 does not correspond to the indicated dimensions.

3) In Table 1, the parameter "Side plate thickness" should be shown in Figure 2.

4) Line 309, page 11. Replace "..in the table below" with "... in the table 2".

5) Line 341, page 11: "In addition, from Table 3, the average crystallite size was found to be 8.16 nm." However, all the values shown in table 3 are much higher (30.92-132.80 nm). Authors should correct this error or they should explain this anomaly.

6) In Figure 12, what is the gas inlet temperature? I have not found that data in the text.

Author Response

The authors are very much thankful to the respected reviewer for accepting to review our manuscript and sharing his valuable time. We cordially acknowledge the useful comments and recommendations made by the respected reviewer on our manuscript. We have tried to revise the manuscript accordingly and the detailed corrections are listed below point by point. The respected reviewer can find all the modifications and corrections in the revised manuscript written in red text. The authors respond to the respected reviewer queries are as follows:

Reviewer comments:

1) There is some confusion about the as-synthesized samples at different temperatures and the measurement of their properties at those temperatures. For instance, is the nanofluid as-prepared at 50ºC always kept at that temperature, to measure its properties? The authors should explain this point more clearly.

- We thank the respected reviewer for his comment. Kindly note that the temperature of the as-synthesized samples was maintained using a dry bath solid block integrated on top of a hot/cold plate device, after which the sample was placed inside it during the measurement to prevent temperature losses. WE have included this detail in the revised version of the manuscript. Kindly see the red text in Section 2.5. Please check the following link, which shows the used dray bath solid block that we have used: https://thermtest.com/thw-l2

Thank you very much. 

2) Line 131, page 3: "A set of clear glass vials of 40 mm outer diameter, 1.6 mm wall thickness, and 95 mm height were obtained..." However, in Figure 1 we can note that the height of the glass vial is greater than 95 mm. In the photograph of Fig. 5, the height of the glass vials is correct. It seems that the vial that appears in Figure 1 does not correspond to the indicated dimensions.

- We thank the respected reviewer for pointing this out. You are correct sir, we mistakenly used the wrong vial in the image (i.e., 165 mm) instead of the 95 mm. Kindly note that this has been corrected in the revised version of the manuscript. Please check Fig. 1 in the revised manuscript. Thank you very much.

3) In Table 1, the parameter "Side plate thickness" should be shown in Figure 2.

- We thank the respected reviewer for his important suggestion. Kindly note that the side plate thickness has been included in Fig. 2 of the revised version of the manuscript as suggested by the respected reviewer. Thank you very much.

4) Line 309, page 11. Replace "..in the table below" with "... in the table 2".

- We thank the respected reviewer for his suggestion. Please note that the suggested changes in the text was taken into account in the revised version of the manuscript. Thank you very much.

5) Line 341, page 11: "In addition, from Table 3, the average crystallite size was found to be 8.16 nm." However, all the values shown in table 3 are much higher (30.92-132.80 nm). Authors should correct this error or they should explain this anomaly.

- We thank the respected reviewer for his comment and apologies for this mistake. Kindly note that this was corrected in the revised manuscript. The XRD crystallite size should have been taken for the highest peak, which correspond to 132.80 Å. This information was included in the revised manuscript. Kindly also note that Table 3 was removed from the revised manuscript upon the request of the respected Reviewer 1 to shorten this section. Thank you very much for pointing this out sir and we apologies that we had to remove the Table.

6) In Figure 12, what is the gas inlet temperature? I have not found that data in the text.

- We thank the respected reviewer for his question. Kindly note that the initial gas inlet temperature was 172.08°C for Fig. 8 (previously Fig. 12), after which it was increased to 407°C for Fig. 9 (previously Fig. 13). We have included this information in our revised version of the manuscript. Please see the red text above Fig. 8. Thank you very much sir.

Finally, the authors would like to thank the respected reviewer for his time and very useful comments and remarks. Thank you very much.

Round 2

Reviewer 1 Report

Dear authors:

Basically I am satisfied with your reply. In addition, whether the length of this paper is too long can be determined by the editor-in-chief.